# Cellular Responses to Widespread DNA Replication Stress

**DOI:** 10.3390/ijms242316903

**Published:** 2023-11-29

**Authors:** Jac A. Nickoloff, Aruna S. Jaiswal, Neelam Sharma, Elizabeth A. Williamson, Manh T. Tran, Dominic Arris, Ming Yang, Robert Hromas

**Affiliations:** 1Department of Environmental and Radiological Health Sciences, Colorado State University, Ft. Collins, CO 80523, USA; 2Department of Medicine and the Mays Cancer Center, The University of Texas Health Science Center San Antonio, San Antonio, TX 78229, USA; jaiswala@uthscsa.edu (A.S.J.); trant10@uthscsa.edu (M.T.T.); hromas@uthscsa.edu (R.H.)

**Keywords:** DNA damage, replication stress, oxidative DNA damage, genome instability, DNA double-strand breaks, structure-specific nucleases, DNA damage response

## Abstract

Replicative DNA polymerases are blocked by nearly all types of DNA damage. The resulting DNA replication stress threatens genome stability. DNA replication stress is also caused by depletion of nucleotide pools, DNA polymerase inhibitors, and DNA sequences or structures that are difficult to replicate. Replication stress triggers complex cellular responses that include cell cycle arrest, replication fork collapse to one-ended DNA double-strand breaks, induction of DNA repair, and programmed cell death after excessive damage. Replication stress caused by specific structures (e.g., G-rich sequences that form G-quadruplexes) is localized but occurs during the S phase of every cell division. This review focuses on cellular responses to widespread stress such as that caused by random DNA damage, DNA polymerase inhibition/nucleotide pool depletion, and R-loops. Another form of global replication stress is seen in cancer cells and is termed oncogenic stress, reflecting dysregulated replication origin firing and/or replication fork progression. Replication stress responses are often dysregulated in cancer cells, and this too contributes to ongoing genome instability that can drive cancer progression. Nucleases play critical roles in replication stress responses, including MUS81, EEPD1, Metnase, CtIP, MRE11, EXO1, DNA2-BLM, SLX1-SLX4, XPF-ERCC1-SLX4, Artemis, XPG, FEN1, and TATDN2. Several of these nucleases cleave branched DNA structures at stressed replication forks to promote repair and restart of these forks. We recently defined roles for EEPD1 in restarting stressed replication forks after oxidative DNA damage, and for TATDN2 in mitigating replication stress caused by R-loop accumulation in BRCA1-defective cells. We also discuss how insights into biological responses to genome-wide replication stress can inform novel cancer treatment strategies that exploit synthetic lethal relationships among replication stress response factors.

## 1. Introduction

Cells respond to DNA damage by activating complex pathways that are collectively termed the DNA damage response (DDR). The DDR has emerged as a crucial network of damage sensing, signaling, and effector pathways that slow or stop cell cycle progression during each cell cycle phase and promote DNA repair and cell survival. However, when damage is excessive, hyperactivated DDR signaling can trigger programmed cell death by apoptosis or other cell death pathways [1,2,3,4,5]. These responses promote cell survival and genome stability in moderately damaged cells, and they eliminate excessively damaged cells, all of which contribute to tumor suppression. It is therefore not surprising that DDR factors are frequently inactivated in cancer, and DDR defects have been shown to destabilize the genome, leading to oncogene activation and/or tumor suppressor inactivation that drive tumorigenesis [4,6,7,8]. For example, in response to DNA damage, p53 plays critical roles in cell cycle checkpoint arrest, it stimulates many DNA repair pathways, and it regulates apoptosis [9,10,11]. Approximately half of tumors have p53 mutations, including inactivating mutations (revealing p53 as a tumor suppressor), and dominant negative (oncogenic gain-of-function) mutations that alter p53 tetramer activities [12]. More than 400 proteins are implicated in the DDR, but unlike p53, most are mutated in cancers at low frequencies [13,14]. The DDR is truly a double-edged sword: it maintains genome stability and suppresses tumorigenesis, but tumor cells can hijack the DDR to upregulate DNA repair and promote tumor cell survival in response to oncogenic stress and stress induced by genotoxic (DNA-damaging) chemotherapy and radiotherapy [15,16,17,18,19,20,21]. Tumors can also upregulate mutagenic repair pathways and this can promote rapid tumor evolution and therapeutic resistance [22,23,24,25]. The central role of the DDR in cancer etiology and treatment response accounts for the substantial ongoing efforts to describe mechanistic features of DDR factors and their attendant pathways, to identify specific DDR defects in tumors that have diagnostic and/or prognostic value, and to exploit DDR pathways and/or weaknesses due to tumor DDR defects to improve treatment of a broad range of cancers [4,6,26,27,28,29,30,31,32,33,34].

S-phase cells are more sensitive to DNA damage than cells in other cell cycle phases because most DNA lesions block DNA replication, and the consequent replication stress can cause lethal DNA double-strand breaks (DSBs) or genome rearrangements (i.e., dicentric chromosomes). DNA lesions that block replication include damaged bases such as those with open rings, chemical adducts (e.g., alkylated bases), oxidized bases, missing bases (apurinic/apyrimidinic sites), deaminated bases, and UV-induced pyrimidine dimers [35,36]. Most tumor cells divide rapidly and are more sensitive to genotoxins than normal tissues that divide infrequently or not at all, providing a therapeutic window for genotoxic chemotherapy and radiotherapy. Although genotoxic cancer therapeutics pose serious challenges due to associated side effects, these agents remain important treatment options, especially when targeted therapeutics are unavailable or tumors develop therapeutic resistance [37,38,39,40,41]. DNA damage interferes with DNA replication, causing replication forks to stall and triggering DNA replication stress, which activates replication stress response pathways that significantly overlap with DDR response pathways [42,43,44]. Thus, acute or chronic exposures to genotoxic chemicals or radiation cause widespread replication stress. Widespread replication stress can also be induced by DNA polymerase inhibitors, such as aphidicolin, and by reduced nucleotide pools [45,46]. Overactive cell growth pathways in cancer, caused for example by hyperactivated or overexpressed oncogenes, also induce widespread replication stress due to dysregulation of DNA replication timing and progression, a phenomenon termed oncogenic stress [47,48,49]. When RNA is generated during transcription, it can form stable RNA–DNA hybrids, termed R-loops, which can cause replication stress when encountered by replication machinery. Failure to properly process R-loops results in genome instability [50]. We include R-loops in the widespread replication stress category because these sequences comprise roughly 5% of the human genome [51]. Even in the absence of DNA damage or other causes of widespread stress, replication stress is a normal feature of the S phase, particularly in metazoan cells with their large and complex genomes [52]. This is because certain DNA sequences are difficult to replicate, including minisatellite repeat sequences at common fragile sites; inverted repeats that can form double stem-loop (cruciform) structures; CAG_n_ triplet repeats that form hairpin structures; purine-rich repeats (e.g., GAA_n_) that are capable of forming triple helical structures; G-rich DNA that can form stable looped, fold-back G-quadruplex structures; and telomere repeats that form branched structures at each end of linear chromosomes [53,54,55,56] (Figure 1). Such difficult-to-replicate sequences are often mutation hotspots, and they can be associated with translocations or other genome rearrangements in human disease, including neurological and developmental diseases, and cancer [57,58,59,60,61,62,63]. We can thus characterize replication stress in two ways: widespread replication stress caused by DNA damage, inhibition of the replication machinery, or oncogenic stress; and localized replication stress at difficult-to-replicate sequences, which cells must manage during every S phase.

There are a variety of replication stress assays. Some assays monitor genome-wide replication such as nucleotide analog incorporation into DNA following pulse-labeling with readily detected thymidine analogs 5-ethynyl-2′-deoxyuridine (EdU), 5-chloro-2′-deoxyuridine (CldU), or bromo-deoxyuridine (BrdU); monitor phosphorylation/activation of checkpoint proteins Chk1, Chk2, ATR, and RPA; monitor phosphorylated histone H2AX (**γ**-H2AX), a marker of DSBs at collapsed forks; monitor cell cycle progression and cell cycle phase profiles; and monitor downstream effects of replication stress including comet assays that detect strand breaks at collapsed replication forks [64,65,66]. These endpoints can be assayed in cell populations using flow cytometry, or in individual cells using immunofluorescence microscopy to detect replication stress-induced nuclear foci. Checkpoint protein phosphorylation and **γ**-H2AX can also be detected using Western blot, or using proteomic mass spectrometry analysis [67]. Genome-wide replication stress can also be assayed with END-seq, a method that detects DSBs at nucleotide resolution [68,69]. Replication stress and replication fork restart are detected at the single-molecule level using DNA fiber analysis or DNA combing, which typically employ pulse-labeling with EdU, or dual pulse-labeling with EdU prior to stress induction, and CldU after release from stress [65,66,70,71,72]. Recruitment of DDR factors to stressed replication forks is detected using iPOND analysis, a type of chromatin immunoprecipitation assay [65,73,74]. These assays focus on cell responses in the S phase, but replication stress also generates single-stranded DNA (ssDNA) gaps detected beyond the S phase and even into subsequent cell cycles [75].

It is likely that replication stress response systems coevolved with genomic features that result in recurrent replication stress, particularly as genomes increased in size and complexity. Given the fundamental importance of accurate genome replication and segregation to daughter cells, it is not surprising that cells evolved multiple, redundant systems to protect stressed replication forks and to repair/restart stalled or damaged forks via (relatively) error-free mechanisms that promote cell survival and protect genome integrity. In the clinic, we can exploit replication stress to attack cancer cells. However, because all cells, including cancer cells, employ robust, redundant stress response systems to counter endogenous and exogenous threats, our task to selectively kill cancer cells requires a detailed understanding of primary and backup stress response systems. Toward this end, we describe here recent advances in our understanding of fork protection and fork repair/restart mechanisms, with a particular focus on mechanisms operating in response to widespread replication stress.

## 2. DNA Lesion Bypass and Stressed Fork Protection Mechanisms

Cells suffer more than 100,000 DNA lesions per day, and it is estimated that there is a steady state of approximately 10,000 lesions per cell [76]. This DNA damage is largely the result of reactive oxygen species (ROS), including superoxide anions (•O_2_^−^), hydroxyl radicals (•OH), hydroxyl ions (OH^−^), and hydrogen peroxide (H_2_O_2_), as well as reactive nitrogen oxides (•ON). ROS are short-lived but extremely reactive with DNA, generating a wide array of lesions including oxidized bases. Most of these lesions are single-strand (SS) damage, including damaged (e.g., 8-oxo-G) or missing bases, and single-strand breaks (SSBs) [76]. SSBs are generated, for example, as intermediates in base excision repair (BER) and nucleotide excision repair (NER) [77,78], and from failed topoisomerase I reactions [79]. Compared with SS damage, DSBs pose a greater threat to genome stability and cell viability, but spontaneous DSBs occur far less frequently, about 25–50 per day in mammalian cells [76,80,81]. Most spontaneous DSBs result from replication stress, and thus primarily occur in S-phase cells [82,83,84,85]. DSBs may also arise when BER or NER intermediates generate closely opposed SSBs [78,86,87,88]. Ionizing radiation (especially high-mass, high-charge particle radiation) and radiomimetic chemicals (e.g., bleomycin, neocardiostatin) create clustered lesions that frequently result in DSBs [89,90].

To maintain genome integrity, DNA must be replicated completely, but only once per cell cycle [91]. Sophisticated origin licensing systems and regulated assembly of prereplication machinery play critical roles in preventing over-replication and consequent genome instability [92,93,94]. Because of these stringent controls, it is important to prevent disassembly of the replisome when replication forks encounter DNA lesions. The vast majority of DNA lesions block replicative DNA polymerases, and replisome disassembly would require reassembly at non-origin (unlicensed) DNA sequences. Several DNA damage tolerance mechanisms have evolved to bypass blocking lesions [95]. One tolerance mechanism is translesion DNA synthesis (TLS), catalyzed by error-prone TLS polymerases (Figure 2A). There are several TLS polymerase families: Y (Pol η, Pol ι, Pol κ, and REV1), A (Pol θ), and B (Pol ζ) [96,97,98]. The less structurally constrained active sites in TLS polymerases allow for lesion bypass, which promotes timely completion of DNA replication, but it also leads to nucleotide misincorporation. Hence, TLS polymerases incur a greater mutagenic cost than the replicative polymerases Pol δ and Pol ε [97,98,99,100]. Mutations tend to be limited to the immediate vicinity of the DNA lesion because TLS polymerases are not highly processive; they synthesize short patches before being exchanged for the highly processive replicative polymerases. Several mechanisms have been proposed to explain TLS polymerase exchange mediated by the homotrimer proliferating cell nuclear antigen (PCNA), described as ‘dynamic exchange’ and ‘tool belt’ models in which alternative polymerases are recruited to and dissociate from PCNA, or remain bound to each of the three PCNA subunits, respectively. A recent model of the ternary PCNA–REV1–Pol η complex suggests dynamic interchange between tool belt and REV1 bridging architectures [101]. In addition, PCNA monoubiquitination regulates polymerase exchange or ‘rotation’ of the tool belt [99,102,103]. Another lesion bypass mechanism involves repriming downstream of the blocked fork, catalyzed by PRIMPOL and PRIM1 (Figure 2B). Unlike most DNA polymerases, PRIMPOL has dual primase and polymerase activities, and PRIMPOL defects have been linked to genome instability and cancer [104]. A recent study indicates that BRCA2 suppresses SS gap formation caused by PRIMPOL repriming, and that unconstrained replication in BRCA2-defective cells underlies the radio-resistant DNA synthesis phenotype of these cells [105]. Human helicase-like transcription factor (HLTF) is a fork remodeling factor that promotes fork reversal, a fork protection mechanism (see below). It was recently shown that in cells experiencing replication stress, HLTF prevents unrestrained DNA synthesis mediated by PRIMPOL and the TLS polymerase REV1 [106]. Thus, it appears that cells balance timely completion of replication and maintenance of genome stability by coordinating various replication stress responses, including TLS, PRIMPOL repriming, and fork protection via fork reversal.

Repriming ahead of a blocked replication fork results in SS gaps. Although there are mechanisms to fill under-replicated SS gap regions, these regions pose risks to genome stability because they are susceptible to nucleolytic attack [107], and because SS gap filling may involve error-prone TLS polymerases [108,109]. However, there remains significant debate about the roles of specific TLS polymerases in SS gap filling [107]. Two additional lesion bypass mechanisms involve DNA polymerase strand switching [110,111], a mechanism related to homologous recombination (HR; Figure 2C), and passive rescue of a stressed fork by an adjacent replication fork (Figure 2D) [112,113]. Fork rescue by an adjacent fork may require activation of a dormant origin of replication, triggered by ATR- and Chk1-dependent activation of the intra-S checkpoint in response to replication stress [114,115]. The intra-S checkpoint also promotes DNA repair activity, slows or stops progression of active replication forks, and prevents late-origin firing, all of which serve to maintain genome stability by limiting replisome encounters with DNA damage [28,116,117,118,119].

When DNA lesions are not bypassed by any of the above mechanisms, blocked replication forks can reverse to a four-way branched structure that resembles a Holiday junction, often termed a ‘chicken foot’ (Figure 3) [120]. Fork reversal serves as a fork protection mechanism, but chicken foot structures have a single-ended DSB (seDSB) that poses risks to genome stability because such ends may be rejoined to a distant DSB (either one end of a two-ended, frank DSB, or a seDSB at another reversed or collapsed fork). In addition, seDSBs are subject to degradation by MRE11 nuclease, a member of the MRE11-RAD50-NBS1 (MRN) complex that plays an important role in DSB detection/early DDR signaling, end-processing, and limited end resection during canonical nonhomologous end joining (cNHEJ) [121].

Fork reversal is promoted by several ATP-dependent fork remodeling motor proteins including HLTF, SMARCAL1, ZRANB3, FBH1, and RAD54L [106,122,123,124,125,126,127,128]. FBH1 helicase also has an F-box domain that acts as part of a Cullin-dependent ubiquitin ligase to regulate replication fork stability. FBH1 helicase can reverse model replication forks and disrupt RAD51 filaments, while the ubiquitin ligase targets RAD51, altering RAD51 levels on chromatin [129,130,131,132,133]. Reversed forks are protected by HR proteins including BRCA1, BRCA2, RAD51, and FANCD2 [134,135,136,137]; the RAD51 regulator RADX [138]; the RIF1 end resection inhibition factor [139]; MRNIP, which interacts with MRN [140]; and USP1, a TLS suppressor that modulates PCNA activity via de-ubiquitination [141]. Cells are hypersensitive to replication stress if any of these fork protection factors are defective or absent. The seDSBs at chicken foot structures are bound by the HR proteins noted above, as well as several RAD51 paralogs (XRCC2, XRCC3, RAD51B, RAD51C, RAD51D) with important roles in HR [142,143]. There is evidence that these HR factors operate in HR and fork protection via distinct mechanisms [144]. In addition to protection from MRE11 nucleolytic attack, the HR proteins RAD51, BRCA1, and BRCA2 also prevent nucleolytic attack of reversed forks by DNA2, MUS81, and EXO1 [126,127,145]. The WRN interacting protein, WRNIP, has also been reported to protect reversed forks from degradation by nucleases [146].

Blocked forks may reverse to different extents. Initial, limited reversal is mediated by several helicases that were first described for their functions in chromatin remodeling, including HLTF, PICH, and SMARCAL1 [120]. Limited fork reversal is also assisted by RAD51, and ZRANB3, an ATP-dependent, structure-specific nuclease, helicase, and strand annealing protein [120,147]. As expected, fork reversal induces topological strain, and the strain induced by extensive fork reversal is relieved by TopoIIα [148]. PICH helicase is recruited to reversed forks by TopoIIα in a mechanism regulated by ZATT-mediated SUMOylation of TopoIIα, and PICH can branch migrate and thereby extend fork reversal [120,148]. Chromatin modification and remodeling factors also protect stressed replication forks. For example, EZH2 methylates histones at blocked replication forks, suppressing MUS81 nuclease recruitment to these forks, and hence providing another layer of fork protection [149]. Forks that undergo limited reversal may restart via RECQ1-mediated branch migration, which restores the normal replication fork structure [150] (Figure 3). The seDSB at reversed forks is rapidly degraded in cells with defects in fork protection factors, accounting for their hypersensitivity to a broad range of replication stress agents. It has been suggested that extensive fork reversal is important for fork restart via HR [120,148]. HR requires significant end resection, which allows the binding of RPA to ssDNA and its subsequent replacement by RAD51 [121,151]. Resected DNA ends suppress cNHEJ [152], thus resection of extensively reversed forks both promotes accurate fork restart mediated by HR and suppresses inappropriate joining of seDSBs at reversed forks and consequent genome instability.

## 3. Replication Fork Restart via Fork Cleavage by Structure-Specific Nucleases MUS81 and EEPD1

The first structure-specific nuclease implicated in replication fork restart was MUS81, an ancient nuclease related to the 3′ XPF endonuclease. First identified in yeast as a UV and methylmethane sulfonate resistance factor, yeast Mus81 was shown to interact with the Rad54 HR protein and function in meiotic HR [153]. In yeast, Mus81 forms a complex with Eme1 (a structure-specific, essential meiosis endonuclease) that can resolve four-way Holiday junctions in vitro [154]. Human MUS81 has similar activity, along with 3′ flap cleavage activity [155]. Human MUS81-EME1 has critical roles in resolving Holiday junctions during HR, and the analogous branched structures at reversed replication forks [156,157,158,159,160]. In human cells, MUS81 forms a complex with another cofactor, EME2, which cleaves blocked replication forks, resulting in fork collapse and a seDSB [161,162,163]. Restart of these collapsed forks involves end resection, RAD51 loading, and strand invasion/extension to reestablish the replication fork (Figure 4), a mechanism analogous to break-induced replication [164]. Yeast Mus81 has also been implicated in the resolution of branched structures when a stalled replication fork is rescued by an adjacent fork [165]. MUS81 has been investigated as a potential cancer therapeutic target, in part because MUS81 defects sensitize cells to genotoxic agents [166,167]. Beyond the obvious use of MUS81 inhibitors to sensitize cells to chemo- or radiotherapy-induced replication stress, MUS81 inhibition is synthetically lethal with the PARP inhibitor olaparib [168]. In this respect, MUS81 defects appear to phenocopy the synthetic lethality of olaparib with BRCA1/2- and other HR-defects [31,169,170,171,172,173,174,175]. These findings suggest a novel way to use olaparib or other PARP inhibitors to treat HR-proficient tumors [168]. Indeed, MUS81 inhibition is synthetically lethal with PARP inhibition [176]. Because BRCA2 helps protect reversed replication forks, cells with BRCA2 defects show enhanced degradation of reversed forks, due to MRE11 attack. In these cells, MUS81 plays an essential role in cleaving blocked forks to allow fork restart by HR [126]. Thus, MUS81 inhibition may serve as an alternative means to sensitize tumors to replication stress induced via genotoxic chemotherapeutics or radiation in BRCA2-defective tumors. On the other hand, functional MUS81 is required for the synthetic lethality seen in cells with defects in the RecQ helicase WRN and mismatch repair [177]. In this case, defective mismatch repair confers microsatellite instability, and expansion of TA dinucleotide repeats causes replication stress due to formation of non-B DNA structures that require unwinding by the WRN helicase. In the absence of WRN, MUS81 cleaves stressed replication forks, leading to massive DSB induction, chromosome shattering, and apoptotic cell death [177]. Therefore, treatment of mismatch repair-defective tumors with inhibitors of both WRN and MUS81 would be counterproductive as it would block MUS81-dependent tumor cell killing.

EEPD1 is a 5′ nuclease with a DNase I-like nuclease domain and a DNA binding helix–hairpin–helix motif related to the RuvA2 helicase [65,178]. RuvA2 is the mammalian homolog to prokaryotic RuvA that binds four-way Holiday junctions [179]. Prokaryotic RuvABC binds and resolves Holiday junctions, and its activities are critical for the rescue of stalled replication forks at UV and other DNA lesions [180,181]. Similarly, EEPD1 promotes replication fork restart in response to replication stress induced by hydroxyurea [65,178], which induces stress by depleting nucleotide pools and by ROS generation [46,182]. siRNA depletion or CRISPR knockout of EEPD1 sensitizes cells to replication stress that causes mitotic catastrophe and chromosome aberrations [65,178]. In response to oxidative stress, EEPD1 is recruited to stressed forks, as revealed via iPOND analysis, and it cleaves replication fork structures in vitro, and stressed replication forks in vivo (Figure 4) [65,178]. As noted above, seDSBs pose risks of genome rearrangement if they rejoin by cNHEJ to ends elsewhere in the genome. To prevent this, EEPD1 recruits EXO1, which resects the end in preparation for fork restart by HR [183]. EEPD1 also promotes resection at frank DSBs, and together these findings reveal EEPD1 as an important upstream regulator of ATR signaling to **γ**-H2AX and Chk1 through RPA-bound ssDNA, and an important HR factor at both frank DSBs and seDSBs at collapsed replication forks [65]. Replication stress is increased in cells undergoing rapid cell division during early embryonic development and in cancer (oncogenic stress), and EEPD1 deficiency causes severe embryonic developmental defects [184].

Because EEPD1 and MUS81 are 5′ and 3′ nucleases, respectively, they cleave stalled replication forks with different polarities (Figure 4). EEPD1 arose later than MUS81, about 450 million years ago during late chordate/early vertebrate evolution [185,186]. This corresponds to the large increase in genome size and complexity [187], which likely increased intrinsic replication stress. As we discussed previously [188], 3′ fork cleavage by MUS81 generates an end that once resected and coated with RAD51 must invade the lagging strand duplex, whereas 5′ cleavage by EEPD1 generates an end that can invade the leading strand duplex (Figure 4). Because the lagging strand comprises discontinuous Okazaki fragments near the replication fork, strand invasion into the lagging strand may be less efficient or perhaps delayed until Okazaki fragment processing is completed. Since even minor delays in fork restart are associated with hypersensitivity to replication stress [65,184,189,190], the potentially faster strand invasion by EEPD1-generated ends may speed fork restart, preventing fork remodeling into toxic HR intermediates and consequent genome instability and cell death [112,191]. Together these findings suggest that EEPD1 provides an alternative to MUS81 to promote fork restart via fork cleavage, one that is potentially faster and with enhanced EXO1 resection to ensure accurate, HR-mediated fork restart [183].

Oxidative DNA damage is very common, and the short-patch BER pathway is critical for reducing oxidative lesion load and replication stress. BER initiates when a member of the DNA glycosylase family cleaves the damaged base, yielding an apurinic/apyrimidinic (AP) site that is processed by AP endonuclease 1 (APE1) to an SSB with 5′-deoxyribosephosphate (dRP) and 3′-hydroxyl ends. Pol β has two activities, a lyase that cleaves the dRP residue, and DNA polymerase that fills the short gap; the nick is sealed by DNA ligase IIIα/XRCC1 [192]. Like TLS polymerases, Pol β is error-prone, but its mutagenic properties are outweighed by its benefits in BER. Defects in Pol β are associated with genome instability and cancer predisposition, and in limited studies Pol β defects are common in colorectal cancer [193,194,195]. We recently described a novel function for EEPD1 in response to oxidative replication stress [72]. EEPD1 is recruited to replication forks blocked by oxidative DNA lesions induced by H_2_O_2_, and it also promotes restart of these stressed forks. Interestingly, EEPD1 promotes resolution of the common 8-oxo-G lesions induced by H_2_O_2_, including those at blocked replication forks, and it also prevents replication fork fusion and fork degradation and thereby limits oxidative damage-induced genome instability [72].

The genome instability associated with EEPD1 defects suggests that EEPD1 mutations might be found in cancers; however, these have not been detected [65], perhaps because the loss of EEPD1 is too destabilizing to sustain rapid cancer cell division in the face of oncogenic stress, nutrient deprivation, hypoxia, and ROS associated with immune inflammation. However, EEPD1 is overexpressed in a wide variety of solid tumor types [196], and this pattern of no inactivating mutations but occasional overexpression is similar to that seen with RAD51, the central HR protein. Thus, EEPD1 (and RAD51) overexpression likely confers a selective advantage to cancer cells as they manage multiple types of stress, and it is also likely to confer resistance to replication stress-inducing cancer chemo- and radiotherapeutics [197]. We recently described an important relationship between BRCA2, RAD52, and EEPD1. As noted above, PARP inhibitors are synthetically lethal with BRCA2 defects, and interestingly, RAD52 defects are also synthetically lethal with BRCA1/2 defects [198,199]. We investigated the RAD52-BRCA1 synthetic lethality and found that it depends on functional EEPD1 [200]. This suggests that EEPD1-mediated HR at stalled replication forks (and possibly frank DSBs) generates intermediates that lead to cell death unless they are processed by BRCA1 and/or RAD52. It further suggests that while RAD52 inhibitors may prove effective in treating cancers with BRCA1 defects, resistance may arise if tumor cells suppress EEPD1 expression or otherwise inactivate EEPD1.

## 4. Other Nucleases Involved in the Replication Stress Response

A large number of nucleases are implicated in replication fork restart and other aspects of cellular stress responses. Because HR-mediated fork restart is required to prevent improper rejoining of seDSBs at collapsed forks, nucleases with various HR functions are important stress response factors. For example, resection is a key early HR step, and CtIP has a well-established role with MRE11 in initiating end resection [151,201]. CtIP also protects reversed forks from DNA2 nuclease degradation, a role that is independent of both MRE11 and CtIP nuclease activity [202]. Because cells with BRCA1 defects display reduced fork protection, CtIP may prove to be an effective target to enhance genotoxic cancer therapies in BRCA1-mutant cancers [202]. The more extensive resection required for HR is mediated by EXO1 and DNA2/BLM [201], and as noted, EEPD1 interaction with EXO1 promotes EXO1-dependent resection at EEPD1-cleaved forks [183]. Metnase (also called SETMAR) is another structure-specific nuclease that promotes restart of stressed replication forks, although it does not directly cleave forks, raising the possibility that it processes branched intermediates at a later stage in the fork restart process [71,178]. Interestingly, Metnase also recruits EXO1 to stressed replication forks, presumably to promote accurate, HR-mediated fork restart [203]. The SLX4 scaffold protein interacts with three structure-specific nucleases, including MUS81, XPF-ERCC1, and SLX1. Although these nucleases play important roles in many different DNA transactions involving branched substrates (e.g., HR, NER, telomere maintenance, interstrand crosslink repair) [204,205,206], there is as yet little direct evidence for their involvement in replication stress responses. The nucleotide excision repair nuclease XPF was recently implicated in the cleavage of stressed replication forks, but the effects of an XPF defect on fork restart speed and overall efficiency were relatively small [207]. XPG, another NER nuclease, has a nuclease-independent role in stressed fork restart that promotes HR and involves interactions with several HR proteins (RAD51, BRCA1/2, PALB2) [208]. Flap endonuclease 1 (FEN1) processes flaps that arise during HR and Okazaki fragments; it has an essential role in removal of flap structures in long-patch BER, and it processes stressed replication forks [209,210,211].

## 5. TATDN2 Is a Structure-Specific RNA Nuclease That Degrades RNA in R-Loops

R-loops typically form during RNA transcription and are a common and widespread source of replication stress. R-loops form when RNA stably pairs with the complementary DNA template strand, displacing the other DNA strand, creating a DNA bubble and RNA–DNA hybrid in a triple-stranded structure [212,213,214]. R-loops are evolutionarily conserved, and play important roles in chromatin remodeling, regulation of transcription, B-cell class switch recombination, RNA-mediated HR, and mitochondrial DNA replication [215,216,217,218,219,220,221,222]. R-loops frequently occur at promoter sequences, transcription termini, and enhancer and super-enhancer elements [213,214,215,216,223,224,225,226,227], and they also mediate recruitment of several factors that regulate histone modifications [214,226,227,228,229]. A critical normal function of R-loops is to relieve replication- or transcription-induced topological stress and, further, topoisomerase defects correlate with increased R-loop formation [230,231,232,233,234,235,236]. Despite these positive roles, R-loops also pose threats, including induction of DSBs, genome instability, and cancer, effects that are likely to reflect their propensity to induce replication stress [214,223,224,237,238,239,240,241,242,243]. One proposal is that replication fork encounters with stable R-loops cause fork collapse to a seDSB leading to fork degradation, fork fusion, and chromosome translocations [241,242,243]. Other, not mutually exclusive, proposals are that the ssDNA in the R-loop DNA bubble is subject to nucleolytic attack [239], or that nucleases involved in transcription-coupled DNA repair cleave R-loop bubbles to induce DSBs [244].

Many proteins modulate R-loop formation and resolution. In yeast, the THO and THSC complexes prevent R-loop formation [245,246]. R-loops can be resolved by RNA unwinding by the helicases senataxin, aquarius (AQR), Werner syndrome protein (WRN), Bloom syndrome protein (BLM), regulator of telomere elongation helicase 1 (RTEL1), petite integration factor (PIF1), Fanconi anemia complementation group M (FANCM), alpha-thalassemia/mental retardation, X-linked (ATRX), and CRISPR-associated DinG protein (CasDinG) [247,248,249,250]. Several DEAD/H-box proteins including DDX1, DDX17, and DHX9 (RNA helicase A) are involved in both R-loop formation and resolution [251,252,253]. Proteins involved in HR and interstrand crosslink repair (FANCA, FANCD2) also function in R-loop resolution [254,255,256,257], and BRCA1 recruits senataxin to R-loops [219,258,259,260,261]. R-loops may also be resolved via RNA degradation by RNases, including RNaseH1 and RNaseH2 [262,263,264,265]. Deficiencies in R-loop resolution are pathological, evidenced by the several human diseases associated with defects in R-loop resolution factors, including amyotrophic lateral sclerosis type 4 and ataxia with oculomotor apraxia type 2 (senataxin) [247,248,249]; Fanconi anemia and various cancers (FANC proteins, BRCA1) [266]; adult-onset mitochondrial encephalomyopathy (RNase H1) [267]; and Aicardi–Goutières neurological syndrome (RNaseH2) [264].

We recently investigated R-loop resolution in BRCA1-mutant cancer cells and found that these cells repress many microRNAs (miRs), probably due to overexpression of IRE1 RNase which degrades miRs involved in tumor suppression [268,269]. Re-expression of one of these miRs, miR-4638-5p, caused BRCA1-mutant cell death, but miR-4638-5p expression did not affect the viability of BRCA1 wild-type cells [66]. miR-4638-5p was found to repress TATDN2, one of three human paralogs of the conserved bacterial TatD nuclease family [270,271,272,273]. We demonstrated that TATDN2 is a Mg^2+^-dependent, structure-specific 3′ RNA exonuclease and endonuclease that resolves R-loops via specific degradation of R-loop RNA [66] (Figure 5A). Downregulation of TATDN2 in BRCA1-mutant cells causes R-loop accumulation, decreased DNA replication, increased DSBs, genome instability, and cell death. Reconstituting BRCA1 in TATDN2-deficient cells suppressed all of these detrimental phenotypes. Together, these results indicate that the increased levels of R-loops in BRCA1-defective cells require TATDN2 for R-loop resolution to maintain genome integrity and cell viability. These results indicate that defects in both TATDN2 and BRCA1 are synthetically lethal (Figure 5B), which suggests novel therapeutic approaches to treat BRCA1-mutant tumors by targeting TATDN2, i.e., with small molecule inhibitors or via expression of miR-4638-5p [66].

## 6. Concluding Remarks

Pathways activated in cells in response to widespread DNA replication stress are complex, and they display significant redundancy, operating in highly interconnected signaling and effector networks. Augmenting the cytotoxic effects of replication stress to treat cancer may take many directions, such as inhibiting upstream checkpoint factors like ATM and ATR [28,30,274,275,276], directly inhibiting replication fork restart nucleases MUS81, EEPD1, or Metnase [166,167,277], and exploiting synthetic lethal relationships such as MUS81-PARP [168], MUS81-BRCA2 [126], BRCA2-RAD52 [198,199], and TATDN2-BRCA1 [66]. A better understanding of replication stress response pathways will also help us to develop strategies that limit or prevent therapeutic resistance, for example, resistance caused by overexpression of replication stress factors such as EEPD1 and RAD51 [196,278]. Although the replication stress field has advanced considerably in the past decade, our mechanistic understanding of specific pathways remains rather cursory. Identification of targets is an important first step, but to exploit targets for cancer therapy often requires a precise mechanistic understanding about how targets will respond to therapeutic intervention. Given the importance of replication stress in cancer etiology and therapeutic response, continued efforts to identify replication stress response targets and their mechanism of action are well justified.

## Figures and Tables

**Figure 1 ijms-24-16903-f001:**
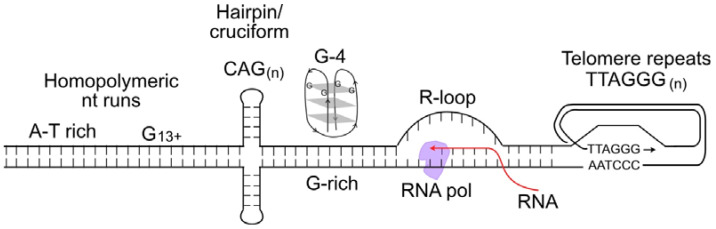
Nucleic acid structures that cause replication stress. These include difficult-to-replicate sequences such as homopolymeric nucleotide runs, palindromes and triplet repeats that can form stem-loop or cruciform structures, G-quadruplex DNA, and self-invading loops at telomeres. Stable R-loops cause replication stress when encountered by replicative DNA polymerases.

**Figure 2 ijms-24-16903-f002:**
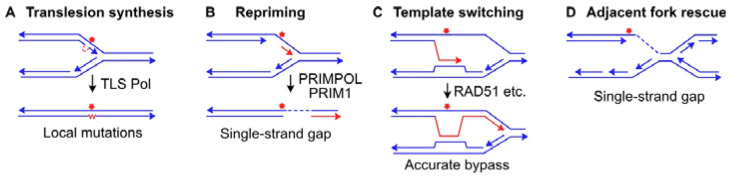
Lesion bypass mechanisms. (**A**) TLS polymerases can synthesize across DNA lesions (red symbol), but with increased mutagenesis. (**B**) Repriming past DNA lesions is accurate but results in SS gaps (dashed line). (**C**) Template switching is an accurate lesion bypass mechanism that involves strand invasion of the sister chromatid for accurate lesion bypass. (**D**) Blocked replication forks can be rescued by an adjacent fork, but similar to repriming, replication is incomplete as it leaves an SS gap.

**Figure 3 ijms-24-16903-f003:**
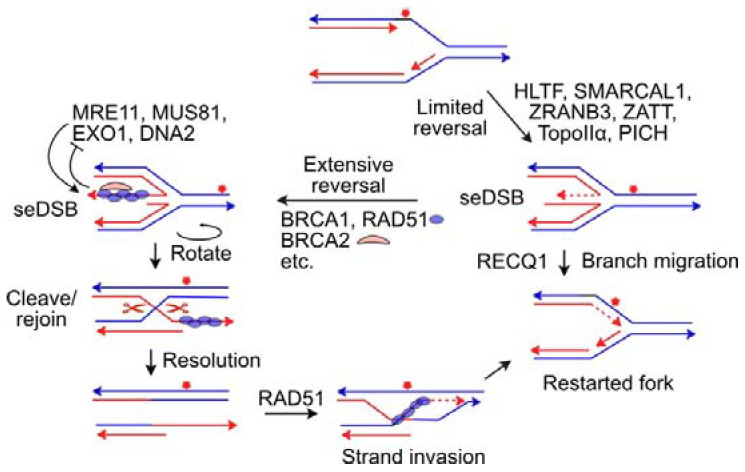
Replication fork reversal, protection, and restart mechanisms. Blocked forks are reversed to a 4-way branched structure (chicken foot) that presents a seDSB. This allows the blocked polymerase to be extended using the nascent sister strand as template (dashed line). Limited fork reversal driven by indicated fork remodeling proteins can be regressed by RECQ1-mediated branch migration to restart the fork. Alternatively, more extensive reversal generates a longer strand at the single-end DSB that is resected and bound by fork protection factors to prevent degradation of the seDSB end by the indicated nucleases.

**Figure 4 ijms-24-16903-f004:**
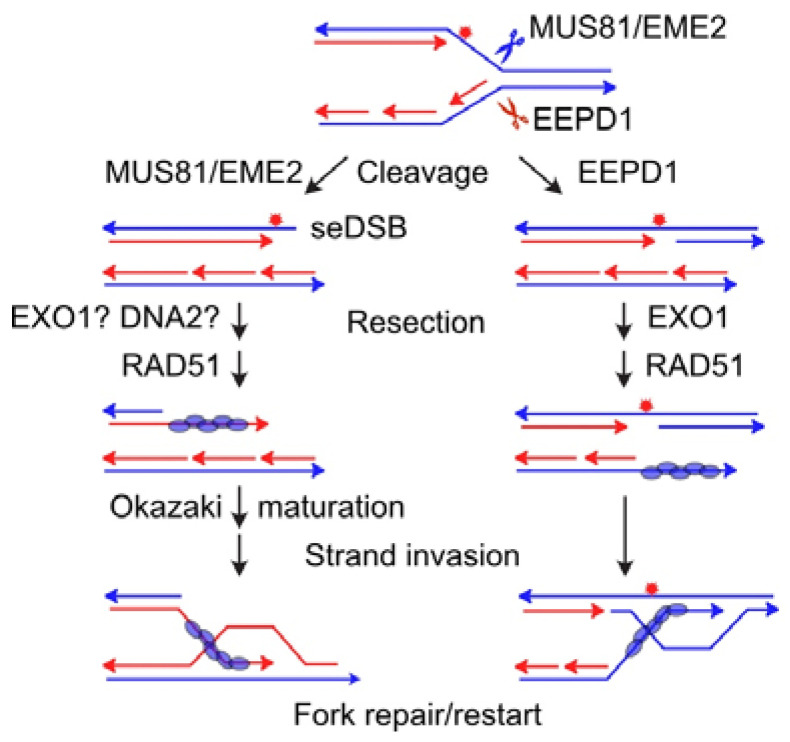
Blocked forks are cleaved by MUS81-EME2 or EEPD1, creating seDSBs. Resection creates ssDNA that is bound by RAD51 to catalyze strand invasion for HR-repair of the broken replication fork. Cleavage by MUS81-EME2 may require extra time for Okazaki fragment maturation before strand invasion.

**Figure 5 ijms-24-16903-f005:**
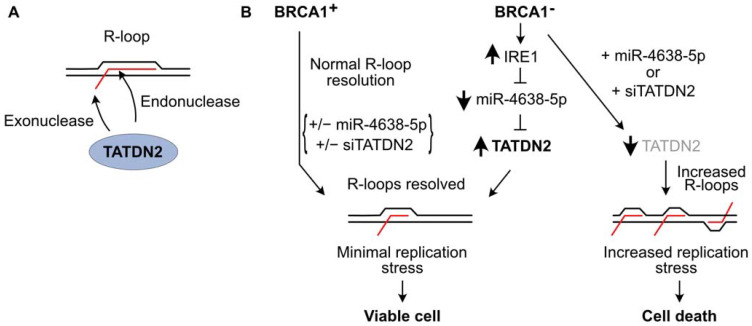
TATDN2 RNase promotes survival of BRCA1-defective cells by suppressing R-loop-induced replication stress. (**A**) TATDN2 is a structure-specific RNase that degrades RNA in R-loops with both exonuclease and endonuclease activities. (**B**) BRCA1 helps cells manage R-loops to prevent toxic replication stress. With functional BRCA1, adding or deleting miR-4638-5p or TATDN2 does not affect cell viability. In BRCA1-deficient cells, IRE1 RNase levels increase, reducing miR-4638-5p and increasing TATDN2 which acts to limit R-loops and associated replication stress, thereby promoting cell viability. Expressing miR-4638-5p or downregulating TATDN2 kills BRCA1-deficient cells due to increased R-loop-associated replication stress.

## Data Availability

All data supporting stated conclusions are presented in the cited references or in the primary literature cited therein.

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
