# Peer review of "Cellular Responses to Widespread DNA Replication Stress"

_ijms, 2023, doi:10.3390/ijms242316903_

Round 1
Reviewer 1 Report
Comments and Suggestions for Authors
The review outlines how DNA damage and other factors can disrupt DNA replication, causing replication stress that threatens genome stability and may lead to cell cycle arrest, DNA repair, or cell death. It emphasizes that this stress is particularly problematic in cancer cells, contributing to genome instability and cancer progression. The role of nucleases in managing replication stress is also highlighted, with a focus on their function in repairing and restarting DNA replication forks.
It is well written and covers major aspects regarding cellular response to replication stress. One suggestion the author may consider is to incorporate a little more background information related to DNA damage in cancer. It seems that the manuscript aims to ends with a special emphasis on DNA damage in tumor development but only providing a few very specific examples. It would be beneficial to expand some points that the authors touch upon, such as:
1. How DNA damage is a double-edged sword in tumorigenesis.
2. Except for TP53, most DDR related genes show low mutation rate but with a long tail, so collectively DDR related genes are indeed frequently mutated in cancers, but each single gene may not stand out.
Author Response
It seems that the manuscript aims to ends with a special emphasis on DNA damage in tumor development but only providing a few very specific examples. It would be beneficial to expand some points that the authors touch upon, such as:
- How DNA damage is a double-edged sword in tumorigenesis.
- Except for TP53, most DDR related genes show low mutation rate but with a long tail, so collectively DDR related genes are indeed frequently mutated in cancers, but each single gene may not stand out.
We thank the reviewers for these excellent suggestions and their time. We expanded these important points in the revised introduction.
Reviewer 2 Report
Comments and Suggestions for Authors
The Review by Nickoloff et al., is well written and organized but presents some weaknesses.
1. The authors should discuss DNA lesions and DNA adducts as a form of replication stress since these represent obstacles perturbing the progression of replication forks.
2. The authors should also consider and briefly discuss the genome-wide assays used to detect different types of replication stress, such as END-seq.
3. The authors mainly focused on the S phase, but replication stress can be spatially and temporarily uncoupled from replication forks. For example, the ssDNA gaps produced during replication may persist into later phases of the cell cycle, even in the next S phase, thus affecting genomic stability in different ways. This is important to be discussed.
Minor comments:
-Pag. 9 lines 437-438: R-loops can be resolved by RNA unwinding by helicases senataxin, aquarius (AQR), Werner syndrome (WRN), Bloom syndrome (BLM)? I believe the authors meant helicases and not syndromes. Please, adjust it.
- Pag 9, line 449: …oculomotor apraxia type 2? Ataxia with oculomotor apraxia type 2.
- Pag. 9 line 455: RNase IRE1 which degrades miRs ... This sentence should read as.. IRE1 RNase which degrades…
Author Response
- The authors should discuss DNA lesions and DNA adducts as a form of replication stress since these represent obstacles perturbing the progression of replication forks.
This was briefly mentioned in the original version. This important point has now been expanded.
- 2. The authors should also consider and briefly discuss the genome-wide assays used to detect different types of replication stress, such as END-seq.
Excellent suggestion – a new paragraph on this topic was added to the introduction.
- The authors mainly focused on the S phase, but replication stress can be spatially and temporarily uncoupled from replication forks. For example, the ssDNA gaps produced during replication may persist into later phases of the cell cycle, even in the next S phase, thus affecting genomic stability in different ways. This is important to be discussed.
This point is included in the new paragraph on stress assays in response to point 2 above.
Minor comments:
-Pag. 9 lines 437-438: R-loops can be resolved by RNA unwinding by helicases senataxin, aquarius (AQR), Werner syndrome (WRN), Bloom syndrome (BLM)? I believe the authors meant helicases and not syndromes. Please, adjust it.
Text revised as suggested:
R-loops can be resolved by RNA unwinding by helicases senataxin, aquarius (AQR), Werner syndrome protein (WRN), Bloom syndrome protein (BLM)…
- Pag 9, line 449: …oculomotor apraxia type 2? Ataxia with oculomotor apraxia type 2.
Corrected.
- Pag. 9 line 455: RNase IRE1 which degrades miRs ... This sentence should read as.. IRE1 RNase which degrades…
Corrected
Round 2
Reviewer 2 Report
Comments and Suggestions for Authors
Excellent Review!